# Maternal Psychopathological Profile during Childbirth and Neonatal Development during the COVID-19 Pandemic: A Pre-Posttest Study

**DOI:** 10.3390/bs13020080

**Published:** 2023-01-18

**Authors:** Sergio Martinez-Vazquez, Blanca Riquelme-Gallego, Leydi Jhoansy Lugo-Toro, Lidia Lucena-Prieto, Nathalia Garrido-Torres, Teresa Lopez-Soto, Rafael A. Caparros-Gonzalez, Alejandro De la Torre-Luque

**Affiliations:** 1Department of Nursing, University of Jaen, 23071 Jaen, Spain; 2Instituto de Investigación Biosanitaria (ibs.GRANADA), 18014 Granada, Spain; 3European University of Valencia, 46010 Valencia, Spain; 4Department of Obstetrics and Gynecology, Delivery Ward, Hospital Antequera, 29200 Malaga, Spain; 5Mental Health Unit, Virgen del Rocio University Hospital, 41013 Seville, Spain; 6English Language Department, University of Seville, 41004 Seville, Spain; 7Department of Nursing, The University of Granada, 18071 Granada, Spain; 8Department of Legal Medicine, Psychiatry and Pathology, Complutense University of Madrid, CIBERSAM ISCIII, 28040 Madrid, Spain

**Keywords:** pregnant women, childbirth, psychopathological profile, coronavirus, COVID-19

## Abstract

The coronavirus SARS-CoV-2 generated an alert that became a state of emergency in health issues worldwide, a situation that affected the entire population, including pregnant women. The present study aims to understand the effect of the psychopathological profile of a sample of pregnant women at the time of the COVID-19 pandemic on themselves during childbirth (Phase 1) and after childbirth and the anthropometric measures of the neonate at birth (Phase 2). The total sample comprises 81 pregnant women aged 32.07 years (SD = 5.45) and their neonates. Sociodemographic and obstetric data of the sample were collected. During pregnancy, psychopathology was measured by means of the SCL-90, as well as other psychological measures on stress and social support. Cluster k-means techniques were used to uncover the heterogeneous profiles of psychopathology in Phase 1. Two main psychopathological profiles were found (Cluster 1: High psychopathological symptoms; Cluster 2: Low psychopathological symptoms). The clusters generated show significant differences in all the SCL-90-R subscales used and in the general index at Phase 1. After childbirth, high psychopathology profile membership was associated with a greater probability of having a non-eutocic delivery. On the other hand, the low psychopathological symptoms cluster shows higher levels of depressive symptoms, hostility, paranoid ideation, and psychotic symptoms in Phase 2. In conclusion, there seemed to exist two heterogeneous profiles of psychopathology in pregnant women during the pandemic; the stress related to the pandemic seemed uninfluential on the development of a profile of high psychopathological symptoms and the psychopathology profile may influence delivery and postpartum outcomes.

## 1. Introduction

In December 2019, a new type of coronavirus, SARS-CoV-2 (COVID-19), was identified in China. The COVID-19 pandemic then quickly spread around the world. In July 2022, more than 575 million cases of COVID-19 were confirmed, including 6.4 million deaths, as reported by the World Health Organization and other sources [1].

Given this context, it may be logical to hypothesize that depression and anxiety may have increased in the population. Due to the uncertainty regarding the spread of the disease, the confinement policies, and even the loss of jobs produced during the pandemic, several factors may have led to the onset of psychological problems across the world [2,3]. COVID-19 has had an impact on people’s mental health in many ways and populations. Mounting studies show that pandemics, natural disasters, and extreme stress may affect pregnant women [4,5].

According to recent data, around 116 million babies will have been born under the shadow of the COVID-19 pandemic, immersed in collapsed health systems as a result of the pandemic, which can interrupt birth control and place the life of the fetus at risk, so an increasing neonatal mortality rate is expected if primary care in hospitals is not reestablished [6]. The trigger of the health emergency in pregnant women plus the sum of the uncertainty may lead to states of elevated depression and anxiety, which is counterproductive during the second and third trimesters of pregnancy [7]. Prenatal maternal stress may also lead to behavioral, emotional, cognitive, and physical alterations, with an evident impact on the neonate’s health [7].

In this sense, the Spanish Marcé Society (2021) confirmed that the prevalence of maternal anxiety disorders during pregnancy has risen by 15.2% [8]. The same phenomenon is observed in the case of maternal perinatal depression, which may reach a prevalence rate of 12.8% during pregnancy. Maternal mental health during a pandemic may have a critical impact on neonatal development [9].

The impact of the pandemic on perinatal mental health has been previously reported in a study by Caparros-Gonzalez, Ganho-Ávila, and de la Torre-Luque [10]. This study focused on examining how exceptional situations such as pandemics or catastrophic events may affect people’s mental health. Pregnant women have been identified as a vulnerable group and are among the most concerned about SARS-CoV-2 spreading and infection. As a result, the authors found that medical visits to hospital decreased for fear of contagion, which may have serious effects on their own physical and mental health. Fear and worries may grow with a subsequent effect on the immune system response of patients [11]. In addition, the health of the fetus may depend on maternal psychological health [11]. Recent studies suggest that the COVID-19 pandemic is associated with a negative psychological impact on pregnant women [12,13].

The impact of a pandemic may have a similar effect of other adverse (or traumatic) events on childbirth and in the fetus [12,13]. The most consistent result is the low birth weight of children who have experienced highly stressful events, with a relationship existing between stress-derived psychopathology and low birth weight [5,11]. Obstetric problems may also be observed during childbirth, such as uterine tears and prolonged deliveries with pregnant mothers exposed to higher stress levels [7]. In a same vein, maternal stress during pregnancy has been associated with the development of neuropsychiatric disorders, such as Autism Spectrum Disorders [9,14]. In this regard, the stress resulting from a concrete situation increased in pregnant women and is related to those cases in which they have suffered from depression and anxiety, and also with low levels of support received from family and friends during the pandemic [3,5,7].

In addition, a recent study reported the psychometric properties of the Pandemic-Related Pregnancy Stress Scale (PREPS) in Spanish-speaking European pregnant women in Spain [9]. The results of this study highlighted the stressful nature that the COVID-19 pandemic impact may have on pregnant women. In this way, congenital effects were analyzed, along with the intrapartum and postnatal product of SARS-CoV-2 infections in pregnant women and their effects on the newborn [11]. This study confirms the postpartum impact of SARS-CoV-2 infections on both mothers and the neonatal fetuses. As a take-home message from the study, preventive measures such as psychotherapy (e.g., online cognitive behavioral intervention) ought to be taken [15].

Besides, in another study on maternal and neonatal consequences of COVID-19 and the infection they may have during pregnancy, the infections caused pneumonia in some cases in pregnant women, but the newborns were more affected than the mother because of their immune system immaturity [10]. Furthermore, the effects of the COVID-19 pandemic on perinatal mental health and the health of the offspring were analyzed. The results showed that the risks of physical and mental health problems may have considerably grown during the pandemic, from respiratory diseases to psychological disorders such as depression and anxiety, for both pregnant women and their babies [16]. Other authors also found an increase in mental health problems among pregnant women in pandemics in general [17].

Therefore, problems in childbirth and later life could be manifested in the fetus because of psychological disorders in the pregnant mother caused by the pandemic. Among the most common consequences on the newborn are the following: affectation of the child’s brain development that can lead to different types of disorders or physical problems; behavioral problems associated with aggressive behavior in some cases; and physical problems in the child that can affect healthy development. Psychological disorders in the pregnant mother may also alter the development of the fetus, which can lead to various pathologies and biochemical dysregulation, which, in turn, may lead to disease development and premature birth [18,19,20,21].

On the other hand, psychopathology symptoms may be associated with physiological alterations of varying biological systems during pregnancy, such as dysregulated stress responses. In this line, it has been shown that the agents of the stress response (CRH or cortisol) may decisively contribute to normative (or diverted) development of the fetus in the prenatal stage [22]. Prenatal stress is a factor involved in the prenatal programming of the disease throughout life, and its influence seems to be exerted in the adult life of the offspring through direct mechanisms (e.g., reduced glucocorticoid resistance) and indirect mechanisms (dysregulation of the inflammatory response) of prenatal programming [14,18,19,20]. Hence, there is a need to screen this population and to correctly characterize those women exposed to postnatal stress with a high risk of suffering mental health problems during pregnancy and in the postpartum period. In this way, it will be possible to make a correct prevention in subsequent prenatal check-ups [23]. Thus, the aim of this study is to understand the effect of the psychopathological profile of a sample of pregnant women at the time of the COVID-19 pandemic on themselves during childbirth and after childbirth and the anthropometric measures of the neonate at birth in order to study the influence on delivery and postpartum outcomes of a profile of high psychopathological symptoms and the psychopathology profile.

## 2. Materials and Methods

A 2-wave study was conducted in a previously reported sample of pregnant women from the South of Spain [9]. The study comprised two measurement points (baseline or Phase 1: between 15 April and 15 May 2020, when mothers were pregnant; and Phase 2, between 6–8 weeks postpartum). The selection of participants was based on the following inclusion criteria: women aged 18 or older, absence of medical diseases, singleton pregnancy, and proficiency in the Spanish language. All participants agreed to collaborate by providing written informed consent. All women were recruited by a midwife during an antenatal appointment. All the study protocols were approved by the Andalusian Biomedical Research Ethics Committee (0904-N-20). The confidentiality of personal information was guaranteed under Spanish Organic Law 3/2018 of December 5th on the Protection of Personal Data. In addition, the study strictly followed the guidelines outlined by the Helsinki Declaration (AMM, 2013) and the Good Clinical Practice Directive (Directive 2005/28/EC) of the European Union.

### 2.1. Instruments

We collected sociodemographic and obstetric data. Moreover, psychopathology symptoms and related factors were collected using the following scales:(a).The Symptom Checklist-90 Revised (SCL-90-R) was used to assess psychopathology symptomatology [24]. The SCL-90-R is a valid and reliable instrument (Cronbach’s alpha = 0.72 to 0.86 across subscales) [25,26]. It is an easy-to-administer inventory that can be self-reported on a wide age range (13 to 65 years), with 90 items on a 5-point Likert scale of response from 1 (not at all) to 5 (very much). It can be used in both the community and clinical settings. The scale evaluates nine symptomatic dimensions: somatization, obsessive-compulsive disorder, interpersonal sensitivity, depression, anxiety, hostility, phobic anxiety, paranoid ideation, and psychoticism. In addition, the SCL-90-R includes seven items that are not incorporated into the nine dimensions but have clinical relevance. Three general indices are constructed from these items: (1) The Global Severity Index (GSI) is an indicator of the current degree of severity of distress; (2) Positive Discomfort Index (PSDI), which indicates self-perception; and (3) Total positive symptoms (TP) to evaluate the response style indicating whether the person tends to exaggerate or minimize the discomfort that afflicts them. PSDI was used in the present study.(b).The Perceived Stress Scale (PSS-14) is a self-administered tool and aims to assess the level of perceived stress in the previous month [27]. The instrument is made up of 14 items on a 5-point Likert scale of response (0 = never, 1 = hardly ever, 2 = occasionally, 3 = often, 4 = very often). A higher PSS total score corresponds to a higher level of perceived stress. This scale was adapted in Spain by Remor and has a reliability of 0.95 Cronbach’s alpha [28].(c).The 12-item Prenatal Distress Questionnaire (PDQ) was also used to assess specific concerns and worries during pregnancy. Responses are on a 5-point Likert scale from 0 (Not at all) to 4 (Extremely) and are summed, providing a total prenatal distress score ranging from 0 to 48. It is easy to administer with a reliability of 0.75 Cronbach’s alpha [29].(d).The 11-item Duke-UNK-11 Functional Social Support was administered using a 5-point Likert-type response scale, which has a reliability of 0.80 Cronbach’s alpha [30].(e).The Pandemic-Related Pregnancy Stress Scale (PREPS) [9,31]. This scale was used to assess pandemic-related pregnancy stress during the COVID-19 pandemic. The PREPS is composed of 15 items scored with a Likert scale ranging from 1 = Very Little to 5 = Very Much and has a Cronbach’s alpha reliability of 0.80.

Due to the number of instruments used and the implications that may have on the results, diagram 1 (see the Appendix A) is provided to explain those connections.

### 2.2. Procedure

After providing the signed informed consent, sociodemographic and obstetric data were collected though an in-person interview (Phase 1). Questionnaires on psychopathology symptoms and related factors were filled out in this phase. More concretely, the participants received a WhatsApp message including a link to the online questionnaires to be completed: (1) SCL-90-R; (2) PSS; (3) PDQ; (4) UNK-11; and (5) PREPS, first during pregnancy (Phase 1). For those women who did not have access to online resources, a telephone interview was conducted. After delivery, in Phase 2 (at 6–8 weeks postpartum), the SCL-90-R was completed again. Women who did not fill out the questionnaire were contacted by phone in order to fill in the scale with them. Obstetric and neonatal variables were collected from the maternal medical history (gestational age at birth, type of delivery, labor onset, rupture of membranes and sex, weight, length, and head circumference of the neonate, respectively).

### 2.3. Data Analysis

Descriptive statistics were obtained for data exploration (mean and standard deviation for continuous variables and percentages for categorical variables).

To identify psychopathological profiles, the k-means clustering technique was used considering the SCL-90 symptom scales. This technique allows participants to be classified according to a series of variables (domains of psychopathology symptoms in our study) into mutually exclusive classes (groups or clusters) based on the multivariate distance (i.e., Euclidean distance) [32]. The algorithm used in this study (NbClust) included 30 cluster extraction indices [33]. The selected clustering solution was ratified using the medoid partitioning algorithm and a cluster-specific silhouette amplitude coefficient. This coefficient indicates how close the identified clusters are. Values close to 1 in this coefficient would indicate that clusters are better differentiated between each other. Depending on the number of clusters obtained, mean comparison tests for continuous measures (t-test or analysis of variance) and Pearson’s χ^2^ test for categorical measures were used to study the relationship between cluster membership and Phase 1 and Phase 2 variables: sociodemographic and clinical variables, and dependent variables during pregnancy and after delivery (psychopathological domains measured by the SCL-90, obstetric variables during delivery and neonate’s anthropometric variables).

The k-means clustering analysis was carried out with the R statistical software (NbClust package) and the rest of the analyses were performed using the IBM SPSS statistical package, version 27.

## 3. Results

The total sample comprised 81 pregnant women with a mean age of 32.07 years old (SD = 5.45) (at phase 1) and their 81 neonates. Sociodemographic and obstetric data are shown in Table 1. A total of 95.1% of the women were married or living with a partner, 3.7% were single-parent families, and 1.2% were divorced. Most women were Spanish-born (81.5%), followed by women born in the South American region (14.8%), European (2.5%), and from Morocco (1.2%). A total of 3.7% had primary education, 53.1% had secondary education, and 43.2% had university studies. A total of 92.6% of the women had a spontaneous pregnancy, 7.4% had become pregnant through an artificial reproductive technique, and 51.9% of them were primiparous. In terms of employment status, 61.7% had a full-time job, 4.9% had a part-time job, and 33.3% were unemployed. A total of 80% had a chronic illness, 18.5% of the sample believed they had been infected with COVID-19, 96.3% had had a positive COVID-19 test, and 97.5% reported having at least one relative with a positive COVID-19 diagnosis.

Obstetric and psychological variables are presented in Table 2. The weeks of pregnancy mean was 36.23, SD = 2.19; the mean number of children was 1.04, SD = 0.24; and the average number of previous abortions suffered was 0.10, SD = 0.40. Regarding the clinical symptomatology, women scored an average of 60.69, SD = 27.07, for somatizations in the SCL-90 scale. Regarding the presence of obsessions, the mean was 64.27, SD = 26.70. The mean of interpersonal sensitivity was 42.40, SD = 1.74. The mean of depression was 50.44, SD = 28.86. The mean of reported anxiety of the sample was 58.64, SD = 30.82, and the mean for feelings of hostility was 49.50, SD = 30.62. The mean phobic anxiety was 64.98, SD = 33.13, and for paranoid ideation, the average was 41,81, SD = 32.21. The mean for psychoticism was 56.25, SD = 34.80. Regarding the Positive Symptom Distress Index, the mean of the sample was 55.17, SD = 28.46, and for the PREPS Scale it was 51.45, SD = 6.50. The PSS mean score was 25.99, SD = 3.74, and the PDQ mean score was 23.83, SD = 3.37. Regarding the Functional Social Support questionnaire, the mean punctuation was 43.22, SD = 6.92.

Cluster analysis revealed that the solution comprising two clusters was the most optimal solution for classifying participants according to the symptomatology dimensions of the SCL-90 during pregnancy. The 2-cluster solution explained 71.9% of the SCL-90 dimension variance. Figure 1 displays how participants were classified into the two clusters. The silhouette amplitude coefficient for cluster 1 (high psychopathology symptom cluster) was 0.43 and for cluster 2 (low psychopathology symptom class) it was 0.30. Cluster 1 comprised 39 women (48.2% of sample), while cluster 2 comprised 42 participants.

None of the differences observed in the sociodemographic and clinical variables were statistically significant between clusters (Appendix A).

Table 3 displays the scores of mothers on psychological variables according to the clusters described above. Significant differences were found across all SCL-90 symptom dimensions and in the general PSDI index (*p* < 0.05). All these scores were significantly higher in cluster 1. No significant between-cluster differences were found for the other scales administered during pregnancy: PSS, PREPS, EEP, PDQ, and Functional Social Support.

Finally, Table 4 shows the data collected in the postpartum period according to the clusters of psychopathological symptoms during pregnancy. Regarding the mother’s mental health, some subscales of the SCL-90 showed significant differences between groups (*p* < 0.05, for all the scales mentioned below). In this regard, scores from four psychopathology dimensions (i.e., depression, hostility, paranoid ideation, and psychoticism) and the PSDI composite scores were significantly higher in women from the low psychopathology cluster in comparison to those from the high psychopathology cluster. Note that the prepartum psychopathology scores from the high psychopathology cluster women were significantly lower than in the postpartum phase. No differences were found for other maternal symptom variables taken after delivery. With respect to obstetric variables, significant differences were observed with respect to the type of delivery: participants from cluster 1 mostly took an instrumental delivery (51.3%), followed by eutocic delivery (30.8%), and caesarean delivery (17.9%), while in cluster 2 there was a higher number of women with a eutocic delivery (78.6%) and a lower number of cases with instrumental (4.8%) and caesarean delivery (16.7%). No significant differences were observed with respect to other obstetric variables, nor were any differences observed with respect to the neonates’ variables.

## 4. Discussion

COVID-19 has had serious implications for the physical and mental health of the population. The present investigation aims to identify the heterogeneous profile of psychopathology symptoms in pregnant women during the COVID-19 pandemic. Moreover, the study analyzes the effect of the psychopathological profile of a sample of pregnant women at the time of the COVID-19 pandemic on themselves during childbirth [34].

It is already known that the perinatal period is a time when women are particularly vulnerable to mental health problems due to morbidity and mortality risk, as observed in pandemics [5,11]. It becomes crucial to gain insight into the actual impact of the pandemic on pregnant women to help implement effective preventive strategies and reduce disease burden. Some existing studies have shown that the development of infections is associated with symptoms of depression and anxiety [2,3,10]. In this vein, it is found that a substantial proportion of pregnant women may show significant anxiety symptoms due to the pandemic-related confinement, probably due to exacerbated feelings of loneliness and lack of social contact [2,3,10].

Although they are often expectant and anxious to await delivery, about 10–20% of pregnant women may also experience mental health problems in the weeks leading up to and following delivery. Varying psychopathology symptoms (e.g., those related to depressive disorder, anxiety, and trauma-related disorders) may therefore arise or be exacerbated [5,11]. Regarding our study, we found that pregnant women may present a well-featured psychopathology symptom profile: either a high psychopathology symptom profile, featured by higher levels of symptoms across the assessed domains, or a low psychopathology symptom profile, featured by lower levels of symptoms. Moreover, the high psychopathology profile members showed higher levels of some types of symptoms postpartum (depression, hostility, paranoid ideation, and psychoticism), even though women from both classes showed similar levels regarding other symptom domains (e.g., anxiety and obsessive-compulsive symptoms).

In terms of delivery outcomes, a relationship was found between psychopathology profile membership and obstetric features of delivery. More concretely, women from the high psychopathology symptom profile show a greater risk of non-eutocic delivery in comparison to women from the low psychopathology symptom profile (OR = 7.92, CI95 = 2.99, 22.89; *p* < 0.01). Our results are in line with existing studies pointing to higher rates of either instrumental or cesarean delivery among women with higher levels of psychopathology symptoms before and during the COVID-19 pandemic [35,36]. Mental health conditions may lead to labor dysfunction and altered uterine contractions as well as a higher need for obstetric interventions [37].

### Strengths and Limitations

Some strengths found in this work are that it is a study in the time of the COVID-19 pandemic with a vulnerable sample of pregnant women identified as a population more at risk to develop mental health issues. Moreover, it constitutes a two-wave study, with two assessment points (during pregnancy and postpartum), which allows exploring possible cause–effect relationships. Finally, it uses scientifically validated measures and instruments and a robust methodological approach. On the other hand, some limitations deserve mentioning. The main limitation is that the study comes from data on an intentional and small sample; in other words, a non-representative Spanish sample, given that the pregnant women were recruited from two southern provinces in Spain (Almeria and Granada). Further studies with a nationally representative sample should be conducted to provide further insight into the effect of the pandemic on the heterogeneity of pregnancy mental health profiles in the time of the COVID-19. On the other hand, some relevant factors were not considered in this study (e.g., history of psychiatric conditions or economic background). Finally, data were only collected when women were in the third trimester of pregnancy, which would not allow for generalizing results across the whole pregnancy. In addition, concerning recall and detection bias, even if they were considered, we do not believe they would affect the results due to the design of the study (two-wave) along with the time of data collection (early after events occurred).

## 5. Conclusions

In conclusion, varying psychopathology symptom profiles emerged in pregnant women in southern Spain during the time of the COVID-19 pandemic, with differential levels of symptoms across psychopathology domains, which may affect delivery and postpartum outcomes. In addition, this study provides some interesting data on the relationship between the woman’s mental health status and fetal development, which may be a line for future research. Moreover, the study may help to develop early detection protocols and treatment for women at risk of elevated mental health symptoms during pregnancy. It becomes crucial for professionals to acquire tools and skills needed to provide high-quality and integrative care to vulnerable women. Based on the results obtained in this research, it is also recommended to carry out further research to provide psychological support protocols for pregnant women.

## Figures and Tables

**Figure 1 behavsci-13-00080-f001:**
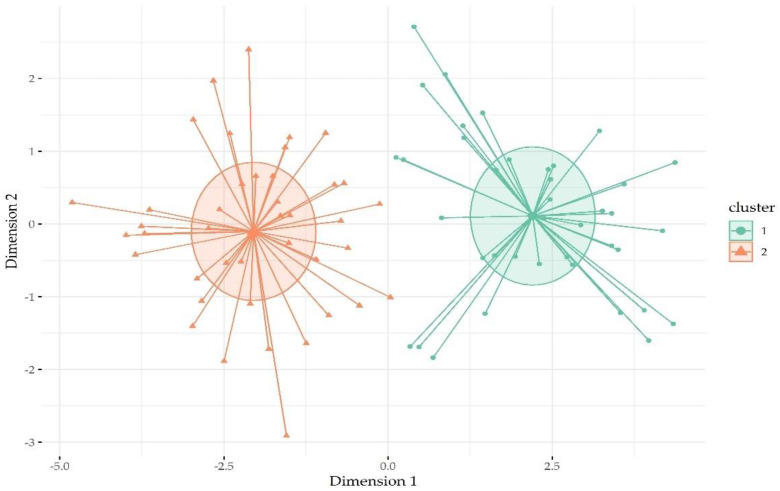
Classification of participants within cluster according to psychopathology symptom distance.

**Table 1 behavsci-13-00080-t001:** Sociodemographic and obstetric data of the sample of women.

	*n*	%
Marital status		
Married or cohabiting	77	95.1
Single	2	3.7
Divorced	1	1.2
Origin		
Spain	66	81.5
South America	12	14.8
Europe	2	2.5
Morocco	1	1.2
Level of studies		
Primary	3	3.7
Secondary	43	53.1
University	35	43.2
Type of pregnancy		
Spontaneous	75	92.6
Assisted reproductive technology	6	7.4
Primiparous (≤1 previous children)		
Yes	42	51.9
Work situation		
Full time	49	61.7
Part time	4	4.9
Unemployed	26	33.3
Previous health problems		
Yes	9	10
No	72	90
Belief of having been infected by COVID-19		
Yes	15	18.5
No	66	81.5
COVID-19 Positive test		
Yes	78	96.3
No	3	3.7
Any relative positive for COVID-19		
Yes	79	97.5
No	2	2.5

**Table 2 behavsci-13-00080-t002:** Obstetric and sociodemographic data and clinical symptomatology of the pregnant women included in the study.

	Mean	SD
Age	32.07	5.45
Gestational age	36.23	2.19
Number of children	1.04	0.24
Previous miscarriages	0.10	0.40
SCL-90-R		
Somatization	60.69	27.07
Obsessive compulsion	64.27	26.70
Interpersonal sensitivity	42.40	31.74
Depression	50.44	28.86
Anxiety	58.64	30.82
Hostility	49.50	30.62
Phobic anxiety	64.98	33.13
Paranoid ideation	41.81	32.21
Psychoticism	56.25	34.80
PSDI	55.17	28.46
PREPS	51.45	6.50
PSS	25.99	3.74
PDQ	23.83	3.37
Functional Social Support	43.22	6.92

SD: Standard deviation; SCL-90-R: Symptom Checklist-90_Revised; PSDI: Positive Symptom Distress Index; PREPS: Pandemic-Related Pregnancy Stress Scale; PSS: Perceived Stress Scale; PDQ: Prenatal Distress Questionnaire.

**Table 3 behavsci-13-00080-t003:** Psychopathological scores of the pregnant women included in the study according to the psychopathological profile cluster.

	**High Psychopathology Cluster**	**Low Psychopathology Cluster**	***t* Test**	** *p* **
SCL-90				
Somatization	80.66 (15.58)	42.14 (21.75)	9.09	<0.05
Obsessive compulsion	83.48 (15.46)	46.42 (22.2)	8.76	<0.05
Interpersonal sensitivity	66.28 (25.14)	20.23 (18.31)	9.36	<0.05
Depression	71.69 (22.44)	30.71 (18.19)	8.98	<0.05
Anxiety	82.05 (19.01)	36.9 (22.65)	9.73	<0.05
Hostility	70.89(19.78)	29.64(25.04)	8.18	<0.05
Phobic anxiety	82.48 (22.7)	48.73 (33.2)	5.37	<0.05
Paranoid ideation	68.89 (21.58)	16.66 (15.72)	12.51	<0.05
Psychoticism	84.66 (14.28)	29.88 (26.35)	11.74	<0.05
PSDI	78.56 (14.58)	33.45 (19.49)	−11.72	<0.05
PREPS	51.05 (6.53)	51.83 (6.53)	−0.53	0.59
PSS	26.21 (4)	25.79 (3.52)	−0.5	0.61
PDQ	24(3.32)	23.67 (3.44)	0.44	0.65
Functional Social Support	43.36 (6.12)	43.09 (7.64)	0.14	0.86

Note. Means and standard deviations for the clusters are presented (in brackets). SCL-90 = Symptom Checklist-90. PSDI = Positive Distress Index. PREPS = Psychometric Properties of the Pandemic-Related Pregnancy Stress Scale. PDQ = The Prenatal Distress Questionnaire.

**Table 4 behavsci-13-00080-t004:** Maternal postpartum psychopathology symptoms and neonate’s anthropometric measures according to the psychopathology profile cluster.

	High Psychopathology Cluster		Low Psychopathology Cluster		Contrast Test	*p*
	Mean	SD	Mean	SD		
Maternal postpartum psychopathological symptoms						
SCL-90-R						
Somatization	47.67	27.57	61.98	27.43	−2.34	0.22
Obsessive compulsion	55.36	29.44	65.12	31.68	−1.43	0.15
Interpersonal sensitivity	42.97	31.39	49.33	36.19	−0.84	0.4
Depression	42.79	30.64	57.5	31.08	−2.14	0.03
Anxiety	51.54	16.31	51.07	15.08	0.13	0.89
Hostility	43.77	32.26	59.98	34.86	−2.16	0.01
Phobic anxiety	62.90	17.83	67.67	18.29	−1.18	0.23
Paranoid ideation	74.74	14.40	83.64	14.36	−278.00	0.00
Psychoticism	44.72	33.83	63.48	33.45	−2.50	0.01
PSDI	50.74	28.37	64.69	31.21	−2.09	0.039
Obstetric data						
Gestational age at birth	38.97	1.51	39.12	1.19	−0.48	0.63
Type of delivery					18.79	0.01
Eutocic	30.8		78.6			
Instrumental	51.3		16.7			
Cesarean	17.9		4.8			
Labor Onset					0	1
Spontaneous	64.1		64.3			
Induced	35.9		35.7			
Rupture of membranes					0.38	0.65
Spontaneous	61.5		54.8			
Artificial	38.5		45.2			
Neonatal anthropometric measures						
Sex					3.58	0.76
Male	61.5		40.5			
Female	38.5		59.5			
Weight (g)	3472.25	379.38	3425.33	432.5	1.08	0.28
Length (cm)	49.85	1.77	50.31	1.26	−1.34	0.18
Head circumference (cm)	33.46	1.86	33.81	1.69	−0.87	0.38

SD: Standard deviation; PSDI: Positive Symptom Distress Index.

## Data Availability

Data are available through the authors at reasonable request.

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
