# Peer review of "Maternal Psychopathological Profile during Childbirth and Neonatal Development during the COVID-19 Pandemic: A Pre-Posttest Study"

_behavsci, 2023, doi:10.3390/bs13020080_

Round 1

Reviewer 1 Report

Greetings and thank you for the opportunity to review "Maternal psychopathological profile during childbirth and neonatal development during the COVID-19 pandemic: A longitudinal study." Although the research is interesting and the findings should be disseminated in the literature, the manuscript requires editing to provide a clear and concise presentation and revisions to address several observed limitations. In this regard, the authors should include the STROBE checklist for reporting cross-sectional studies (https://www.equator-network.org/reporting-guidelines/strobe/) as a supplemental file with the line numbers noted for each reporting element.

Due to my expertise in this area of research as well as extensive experience as a methodologist working with applied psychometrics in cross-cultural research, the report is detailed and direct to specific points. Despite the critical comments and multiple recommendations, the report is intended to be constructive for the purpose of a successful revision process. With this stated, the pdf of the manuscript is attached with some more specific comments related to the report. I look forward to reviewing the revised manuscript upon resubmission. 

----------- Review Report -----------

The introduction is very long hence the authors have probably confused the introduction with what should be called the background as another section. The introduction is usually two to three paragraphs to introduce the topic followed by the structured background information to support the purpose of the study and to justify the significance. The introduction does not describe the theoretical framework or conceptual model where the concepts of interest are stated in the research question and then operationalized into variables for measurement with the instruments presented in the introduction. For this reason, I recommend the authors provide a diagram to clearly explain the hypothesized relationships. This information should align with the five instruments, including the specific dimensions in each instrument utilized for analysis, presented in the methods section.

The authors discuss in the introduction the "...psychopathological profile of a pregnant women..." (lines 124-125) without defining the meaning of this concept and the relevance, or justification, for research. Then, the same statement is used in the study purpose, "Thus, the aim of the study was to understand the effect of the psychopathological profile of a sample of pregnant women at the time of the COVID-19 pandemic on themselves during childbirth, after childbirth and the anthropometric measures of the neonate at birth" (lines 126-129). As a notation, paragraphs don't usually begin with the word "thus" so I think the line should be attached to the prior paragraph as the concluding sentence.

In the materials and methods, the authors stated, "A prospective longitudinal study was conducted in a sample of pregnant women from the South of Spain" (lines 131-132). Then, the study was defined as being two phases, "The study comprised two waves (baseline or Phase 1: between 15 April and 15 May 2020, when mothers were pregnant; and a follow-up of Phase 2, between 6-8 weeks postpartum)" (lines 132-133). The manuscript does not report a longitudinal study in terms of defining the study design. Instead, this study is cross-sectional with two points of measurement, cross-sections, for data collection spaced no more than two months apart. Importantly, I was unable to clearly identify how t1 and t2 data were used. Please provide more information about this point in case I missed something in the manuscript.

The sample for the current study appears to overlap with another published study in Midwifery that I read last year. In this regard, this sentence was identified in the introduction that needs to be clarified "In addition, a recent study reported the psychometric properties of the Pandemic-Related Pregnancy Stress Scale (PREPS) in Spanish-speaking European pregnant women 89 in Spain [15]. The results of this study highlighted the stressful nature that the COVID-19 90 pandemic impact may have on pregnant women" (lines 88-91). The authors need to clearly identify the overlap in the studies in terms of the participants. If the same data set was used, even part of the data set, this needs to be disclosed. Remember, the aim of the study in Midwifery was "to develop and establish the psychometric properties of the Pandemic-Related Pregnancy Stress Scale (PREPS) in European Spanish speaking pregnant women in Spain." For this reason, the rationale for the overlap is quite important to disclose unless my perception of the same database is incorrect.

In the results, the statement "The total sample comprised 81 pregnant women with a mean age of 32.07 years (SD 213 = 5.45) (at phase 1) and their 81 neonates" (lines 213-214) indicates the neonates were participants. However, I am unable to find any measurements or analysis for these participants.

The authors stated, "Cluster analysis revealed that the solution comprising two clusters was the most optimal solution for classifying participants according to the symptomatology dimensions of the SCL-90 during pregnancy" (lines 249-251). The relevance of this finding is unclear given the lack of an a priori framework or model specific to the hypothesized reality. 

Table 3 is a rather large table with lots of data but the relevance is not established as the authors admit, "None of the differences observed in the socio-demographic and clinical variables were statistically significant between clusters" (lines 263-265). For this reason, table 3 should be removed from the manuscript as irrelevant to the findings. However, this could be included in a supplemental file.

The authors stated, "Significant differences were found across all SCL-90 symptom dimensions and in the general PSDI index (p < .05). All these scores were significantly higher in Cluster 1. No significant between-cluster differences were found for the other scales administered in pregnancy: PSS, PREPS, EEP, PDQ and UNK-11" but there is no framework or model presented to understand the significance of these findings in a scientific manner rather than a data mining exercise.  Also, a large amount of the data presented in Table 4 should be eliminated as unnecessary because the findings for only one scale are significant while the others are not. 

Table 5 is mislabeled as Table 6. Again, Table 6 is too long and busy with most of the data not having significance. Please abbreviate all the tables to focus on the important points in the findings. The readers are not willing to sort through the nonsensical data to find the important data.

Discussion needs to be tightened as the findings are one thing but the discussion seems to address other things. A large amount of the discussion is essentially repeating the results in a descriptive manner. Then, there is little to no information presented from other studies.

Please label the area at the end of the discussion as limitations and strengths. Also, please report the limitations for the study first followed by the strengths. The limitations need to address the biases associated with an observational study using instruments. None of these biases are noted in the limitations.

The conclusion is overstated. The authors stated, "In conclusion, varying psychopathology symptom profiles emerged in pregnant women during the times of the COVID-19 pandemic, with differential levels of symptoms across psychopathology domains which directly affected delivery and post-partum outcomes" (lines 377-380). First, the profiles for pregnant women in Southern Spain. Second, the findings did not causally link delivery and postpartum outcomes. As such this information needs to be clarified.

The authors stated, "In addition, this study provides some interesting data on the relationship between the woman’s mental health status and the fetal development, that may be greatly useful  for healthcare professionals involved in pregnancy care, labor, and postpartum" (lines 380-382). This statement is unclear and too vague to be a conclusion.

The authors stated, "the study may help develop early detection protocols and treatment for women at risk of elevated mental health symptoms during pregnancy" (lines 383-384). There is nothing in the discussion to suggest this is appropriate or how this might be completed. Similarly, the authors stated "becomes crucial that professionals to acquire tools and skills needed to provide high-quality and integrative care to vulnerable women" (lines 384-385), but pregnancy is not vulnerable in terms of a description for a population. In the discussion, the tools and skills needed by professionals to be evidenced specific to the findings of this study.

Finally, 10 of the 48 references are self citations (21%). This is an important notation as there are areas in the manuscript with poor referencing. For example, the authors stated with their own editorial, "The impact of the pandemic on perinatal mental health has been previously reported in a studied by Caparros-Gonzalez, Ganho-Ávila and de la Torre-Luque [10]" (lines 66-67). This statement should be primary research sources rather than an editorial.

Author Response

Comments and Suggestions for Authors

Greetings and thank you for the opportunity to review "Maternal psychopathological profile during childbirth and neonatal development during the COVID-19 pandemic: A longitudinal study." Although the research is interesting and the findings should be disseminated in the literature, the manuscript requires editing to provide a clear and concise presentation and revisions to address several observed limitations. In this regard, the authors should include the STROBE checklist for reporting cross-sectional studies (https://www.equator-network.org/reporting-guidelines/strobe/) as a supplemental file with the line numbers noted for each reporting element.

Due to my expertise in this area of research as well as extensive experience as a methodologist working with applied psychometrics in cross-cultural research, the report is detailed and direct to specific points. Despite the critical comments and multiple recommendations, the report is intended to be constructive for the purpose of a successful revision process. With this stated, the pdf of the manuscript is attached with some more specific comments related to the report. I look forward to reviewing the revised manuscript upon resubmission. 

Thank you very much for taking the time to review this work. It is a pleasure to receive your kind comments with the aim of improving the manuscript.

We believe that the recommendations you have given us are completely consistent and have been reflected throughout the manuscript as you will see below.

We also thank you for your specific comments, which have made it possible to correct and expand the manuscript point by point.

Since this is an extensive approach, we believe that the new manuscript will be of greater value, providing higher methodological quality, which will yield more robust results than those previously reported.

The STROBE checklist has been completed as recommended by you. We believe that our study meets virtually all items of these recommendations. However, certain items are noted that have not been addressed, a limitation that we are aware of and we note accordingly in the discussion section.

----------- Review Report -----------

Remark 1: The introduction is very long hence the authors have probably confused the introduction with what should be called the background as another section. The introduction is usually two to three paragraphs to introduce the topic followed by the structured background information to support the purpose of the study and to justify the significance.

Thank you for your comments. We organized the introduction following the instructions for the author's section in the journal criteria (https://www.mdpi.com/journal/behavsci/instructions) this may give the impression as the introduction is long, but as the dear reviewer noted, we included a brief introduction of the topic, structured background to support the purpose of the study and justify why this study should be taken into account. Instead of dividing it into sections, we decided to narratively debrief the contents.

Remark 2: The introduction does not describe the theoretical framework or conceptual model where the concepts of interest are stated in the research question and then operationalized into variables for measurement with the instruments presented in the introduction. For this reason, I recommend the authors provide a diagram to clearly explain the hypothesized relationships. This information should align with the five instruments, including the specific dimensions in each instrument utilized for analysis, presented in the methods section.

Thank you for your comments. Line 184: “Due to the number of instruments used and the implications that may have in the results, diagram 1 is provided to explain those connections.” Is been added in section 2.1 Instruments; as well as diagram 1 as supplementary material.

Remark 3: The authors discuss in the introduction the "...psychopathological profile of a pregnant women..." (lines 124-125) without defining the meaning of this concept and the relevance, or justification, for research. Then, the same statement is used in the study purpose, "Thus, the aim of the study was to understand the effect of the psychopathological profile of a sample of pregnant women at the time of the COVID-19 pandemic on themselves during childbirth, after childbirth and the anthropometric measures of the neonate at birth" (lines 126-129). As a notation, paragraphs don't usually begin with the word "thus" so I think the line should be attached to the prior paragraph as the concluding sentence.

The connectors used were changed as well as provided a more in-depth explanation of the psychopathological profile. Also in line 153, Instrument A), The Symptom Checklist-90 Revised (SCL-90-R), provides the content to contextualize the meaning of this profile based on the subscales and dimensions assessed by the scale.

Remark 4: In the materials and methods, the authors stated, "A prospective longitudinal study was conducted in a sample of pregnant women from the South of Spain" (lines 131-132). Then, the study was defined as being two phases, "The study comprised two waves (baseline or Phase 1: between 15 April and 15 May 2020, when mothers were pregnant; and a follow-up of Phase 2, between 6-8 weeks postpartum)" (lines 132-133). The manuscript does not report a longitudinal study in terms of defining the study design. Instead, this study is cross-sectional with two points of measurement, cross-sections, for data collection spaced no more than two months apart.

We agree with this comment. Despite having collected data during pregnancy and after delivery, the way in which these data have been treated is not consistent with a longitudinal methodology.

This has been conveniently changed in the manuscript in the Method section:

"A cross-sectional prospective longitudinal study was conducted in a sample of pregnant women from the South of Spain. The study comprised two measurement points waves (baseline or Phase 1: between 15 April and 15 May 2020, when mothers were pregnant; and a follow-up of Phase 2, between 6-8 weeks postpartum). " as well as in the title.

We further believe that the following sentence is redundant:

"The study was divided into two phases; the first phase was conducted during the three trimesters of pregnancy and the second phase was after delivery", so it has therefore been removed from the manuscript.

Importantly, I was unable to clearly identify how t1 and t2 data were used. Please provide more information about this point in case I missed something in the manuscript.

We agree that the use of obstetric and neonatal variables has not been specified. These variables collected during the postpartum period were used to be analysed according to clusters of psychopathological symptoms during pregnancy. Although no significant differences were found between the two clusters for most of the variables, we still find it interesting to show these results.

The following sentence has been added in the methods section to explain more clearly how the variables collected in the postpartum period were used:

“Depending on the number of clusters obtained, mean comparison tests for continuous measures (t-test or analysis of variance) and Pearson's x2 test for categorical measures were used to study the relationship between cluster membership and Phase 1 and Phase 2 variables: sociodemographic and clinical variables, and dependent variables during pregnancy and after delivery (psychopathological domains measured by the SCL-90, obstetric variables during delivery and neonate’s anthropometric variables).”

Remark 5: The sample for the current study appears to overlap with another published study in Midwifery that I read last year. In this regard, this sentence was identified in the introduction that needs to be clarified "In addition, a recent study reported the psychometric properties of the Pandemic-Related Pregnancy Stress Scale (PREPS) in Spanish-speaking European pregnant women 89 in Spain [15]. The results of this study highlighted the stressful nature that the COVID-19 90 pandemic impact may have on pregnant women" (lines 88-91). The authors need to clearly identify the overlap in the studies in terms of the participants. If the same data set was used, even part of the data set, this needs to be disclosed. Remember, the aim of the study in Midwifery was "to develop and establish the psychometric properties of the Pandemic-Related Pregnancy Stress Scale (PREPS) in European Spanish speaking pregnant women in Spain." For this reason, the rationale for the overlap is quite important to disclose unless my perception of the same database is incorrect.

The reviewer is right, this is the same sample of women as in the Midwifery study. This has been conveniently indicated in the first sentence of the methods section:

“A cross-sectional study was conducted in a previously reported sample of pregnant women from the South of Spain [15]”.

Garcia-Silva, J.; Caracuel, A.; Lozano-Ruiz, A.; Alderdice, F.; Lobel, M.; Perra, O.; Caparros-Gonzalez, R.A. Pandemic-Related Pregnancy Stress among Pregnant Women during the COVID-19 Pandemic in Spain. Midwifery 2021, 103, 103163, doi:10.1016/j.midw.2021.103163.

Remark 6: In the results, the statement "The total sample comprised 81 pregnant women with a mean age of 32.07 years (SD 213 = 5.45) (at phase 1) and their 81 neonates" (lines 213-214) indicates the neonates were participants. However, I am unable to find any measurements or analysis for these participants.

These results are presented in table 5. They are the difference in means of neonatal anthropometric measures between the two clusters of psychopathological profiles.

The result of this difference in means is also explained in the results (lines 299-300):

"Nor were any differences observed with respect to the neonate's variables. "

Remark 7: The authors stated, "Cluster analysis revealed that the solution comprising two clusters was the most optimal solution for classifying participants according to the symptomatology dimensions of the SCL-90 during pregnancy" (lines 249-251). The relevance of this finding is unclear given the lack of an a priori framework or model specific to the hypothesized reality. 

We agree with this comment.

The introduction mentions the consequences of being exposed to prenatal stress, both on maternal health and on the impact on neonatal and lifelong health of offspring. Thus, it is clear that there is a need to characterize these women as early as possible through screening for the prevention of mental health problems in the postpartum period. However, the authors of the manuscript are aware that we have not sufficiently justified this aspect in the introductory section. The following paragraph has been included in the Introduction section to justify why women have been classified into two clusters according to SCL-90 scores:

The following sentence has been replaced:

“Hence, the importance of knowing the psychopathological profile of pregnant women.

By the following:

“Hence the need to screen this population and to correctly characterise those women exposed to postnatal stress with a high risk of suffering mental health problems during pregnancy and in the postpartum period. In this way, it will be possible to make a correct prevention in subsequent prenatal appointments [24].”

[24] Shaw, S.H.; Herbers, J.E.; Cutuli, J.J. Medical and Psychosocial Risk Profiles for Low Birthweight and Preterm Birth. Women’s Heal. Issues 2019, 29, 400–406, doi:10.1016/j.whi.2019.06.005.

Remark 8: Table 3 is a rather large table with lots of data but the relevance is not established as the authors admit, "None of the differences observed in the socio-demographic and clinical variables were statistically significant between clusters" (lines 263-265). For this reason, table 3 should be removed from the manuscript as irrelevant to the findings. However, this could be included in a supplemental file.

We agree with the reviewer's comments. This table has been included as supplementary material.

Remark 9: The authors stated, "Significant differences were found across all SCL-90 symptom dimensions and in the general PSDI index (p < .05). All these scores were significantly higher in Cluster 1. No significant between-cluster differences were found for the other scales administered in pregnancy: PSS, PREPS, EEP, PDQ and UNK-11" but there is no framework or model presented to understand the significance of these findings in a scientific manner rather than a data mining exercise.  Also, a large amount of the data presented in Table 4 should be eliminated as unnecessary because the findings for only one scale are significant while the others are not. 

We agree with the comment of reviewer 1. We have not fully justified this finding. Since we have introduced the other psychopathological tests into the analysis, we believe that it may be valuable to show these results despite the fact that there are no significant differences between the two profiles identified in the study population.

The authors of the study believe that in our sample, the SCL-90-R was more sensitive in finding differences, perhaps due to the large number of questionnaire items and subscales it contains.  

Remark 10: Table 5 is mislabeled as Table 6. Again, Table 6 is too long and busy with most of the data not having significance. Please abbreviate all the tables to focus on the important points in the findings. The readers are not willing to sort through the nonsensical data to find the important data.

We agree with the reviewer's comments, the numbering of the tables has been corrected accordingly after putting table 3 as supplementary material. Likewise redundant data have been omitted from table 5, which has now become table 4. 

Remark 11: Discussion needs to be tightened as the findings are one thing but the discussion seems to address other things. A large amount of the discussion is essentially repeating the results in a descriptive manner. Then, there is little to no information presented from other studies.

Thank you for your comments. The discussion has been made shorter, and now is more concise. Some of the statements are supported now with references. Including an appreciation of the vulnerability of pregnant women.

Remark 12: Please label the area at the end of the discussion as limitations and strengths. Also, please report the limitations for the study first followed by the strengths. The limitations need to address the biases associated with an observational study using instruments. None of these biases are noted in the limitations.

Thank you for your comments. We separated this section providing more explanation about the possible bias contemplated from the design, data collection, or analysis. We hope those modifications empowered the approach of the study.

Remark 13: The conclusion is overstated. The authors stated, "In conclusion, varying psychopathology symptom profiles emerged in pregnant women during the times of the COVID-19 pandemic, with differential levels of symptoms across psychopathology domains which directly affected delivery and post-partum outcomes" (lines 377-380). First, the profiles for pregnant women in Southern Spain. Second, the findings did not causally link delivery and postpartum outcomes. As such this information needs to be clarified.

Thank you for your comments. We agree with this clarification, as was contextualized in the limitations section, we included the population context and use more cautious language when making connections between statistical results and implications for its meaning in the sample.

Remark 14: The authors stated, "In addition, this study provides some interesting data on the relationship between the woman’s mental health status and the fetal development, that may be greatly useful  for healthcare professionals involved in pregnancy care, labor, and postpartum" (lines 380-382). This statement is unclear and too vague to be a conclusion.

Thank you for your comments. It may seem vague but is more than a statement, a future research line to explore. We saw some statistical differences pointing out in this direction, however, can´t make assumptions with the data available. We agree with this comment, then was approached more from this perspective.

Remark 15: The authors stated, "the study may help develop early detection protocols and treatment for women at risk of elevated mental health symptoms during pregnancy" (lines 383-384). There is nothing in the discussion to suggest this is appropriate or how this might be completed. Similarly, the authors stated "becomes crucial that professionals to acquire tools and skills needed to provide high-quality and integrative care to vulnerable women" (lines 384-385), but pregnancy is not vulnerable in terms of a description for a population. In the discussion, the tools and skills needed by professionals to be evidenced specific to the findings of this study.

Finally, 10 of the 48 references are self citations (21%). This is an important notation as there are areas in the manuscript with poor referencing. For example, the authors stated with their own editorial, "The impact of the pandemic on perinatal mental health has been previously reported in a studied by Caparros-Gonzalez, Ganho-Ávila and de la Torre-Luque [10]" (lines 66-67). This statement should be primary research sources rather than an editorial.

Thank you for your comments. Pregnant women are considered by different authors a vulnerable population, especially for mental health issues (reference added from the previous comment from WHO). This is due to the increased risk of developing those pathologies. Basing our model on this statement and seeking established relations that improve the care given to those women, we assessed them with various instruments. In other words, we became aware of the risk increase and for that reason, our approach was more in-depth.

In the second part of the comment, we do agree with this statement. However, we see it as a strength that empowers the findings presented here: In the first steps of our research, when conducting a literature review, the lack of studies linking those concepts was patent. Then, the opportunity to cooperate with the authors mentioned emerged as a possibility not only to use their vast knowledge, but also its previous findings as a supporting statement for our statistical associations. In other words, is not a case of self-citation, is a case of collaboration merged with a lack of studies focused on this topic which may highlight the novelty and importance of the findings in the present study.

Reviewer 2 Report

The study originality is  elevated, the pandemic context for the fragile pregnant women profile , being in interesting subject.

But.. the icnlusion and exclusion critetia must be more clearlly described, the group comorbidities must be developed, to reduce the bias. Eventually to have a retrospective evaluation (same questionnaire- completed before the pandemy, to know the psychological profile before the pandemy and pregnancy)

I recommend to calculate OR , not only t test,  to have a more accurate statistical analysis.

I consider suitable for the authors, too, the  EQ-5D-3L , another  important health questinnaire  which could be used for the  pregnant women.

Author Response

The study originality is  elevated, the pandemic context for the fragile pregnant women profile , being in interesting subject.

Thank you very much for taking the time to review this work. It is a pleasure to receive your kind comments with the aim of improving the manuscript.

We believe that the recommendations you have given us are completely consistent and have been reflected throughout the manuscript as you will see below.

We also thank you for your specific comments, which have made it possible to correct and expand the manuscript point by point.

But.. Remark 1: the icnlusion and exclusion critetia must be more clearlly described, the group comorbidities must be developed, to reduce the bias.

We agree with this comment. The inclusion and exclusion criteria have not been correctly addressed.

The following paragraph has been added to the methods section:

“The selection of participants based on the following inclusion criteria: women aged 18 or older, spontaneous pregnancy, absence of medical diseases, and a singleton pregnancy, proficient in the Spanish language; and agreeing to participate by providing a written informed consent. All women were recruited by a midwife during an antenatal appointment”.

Remark 2: Eventually to have a retrospective evaluation (same questionnaire- completed before the pandemy, to know the psychological profile before the pandemy and pregnancy)

We understand that it would be very important to know the prevalence before the pandemic to check whether this increase in the scores of the different psychological tests (especially the SCL) was specifically due to the outbreak of the COVID-19 pandemic. There is a large body of evidence that has shown how the prevalence of depressive and anxiety symptoms during pregnancy and in the postnatal period has increased . However, the methodology of the study we present here did not allow us to do so. This limitation is addressed in the discussion section under strengths and limitations.

Remark 3: I recommend to calculate OR , not only t test,  to have a more accurate statistical analysis.

Thank you very much for your comment, the OR is a statistic from a logistic regression analysis that would provide more precise information to answer our research question. However, to perform such an analysis, a larger clinical population would be needed. In addition, although some of the participants in our study had some of the clinical symptoms of the scales and subscales (according to the high and low psychopathology clusters), most of the participants were healthy, so we do not have a sufficient number of participants in our sample to perform this type of analysis. However, we will not hesitate to take this into account for future studies.

Remark 4: I consider suitable for the authors, too, the  EQ-5D-3L , another  important health questinnaire  which could be used for the  pregnant women.

We are grateful to reviewer 2 for recommending the use of the EQ-5D-3L questionnaire, which would yield much more information to be contrasted with the questionnaires used in the present study. We will keep this in mind for future research with a similar objective.

Round 2

Reviewer 2 Report

The number of participants is limited . An extensive analysis could be useful for more strenght conclusion. This is the limitation of the study.

Author Response

31 December 2022

Dear Dr. Li

Managing Editor

Behavioral Sciences

Manuscript ID: behavsci-1934860

Manuscript title: Maternal psychopathological profile during childbirth and neonatal development during the COVID-19 pandemic: a longitudinal study

Dear Dr. Li,

Thank you for your letter dated 28 December 2022. We are pleased to know that the reviewers and editor recommended reconsideration of our paper following minor revision. We would like to thank this editor for inviting us to resubmit our manuscript to Behavioral Sciences after addressing all reviewers´ and editor´s comments.

We have modified the paper in response to the reviewer comments. I am therefore resubmitting the paper with the suggested revisions (using the track-changes mode in the manuscript).

Yours sincerely,

Blanca Riquelme-Gallego     

# Reviewer 1

The number of participants is limited . An extensive analysis could be useful for more strength conclusion. This is the limitation of the study.

We would like to thank you for your comment. Indeed, this is not a very large sample of the population and, therefore, it is a limitation to consider when extrapolating the results of this research to the rest of the population.

This limitation has been added to the discussion section (lines 385-386).

With respect to the analyses, the authors of the article believe that they are adequate given the data collected. However, we will take this into account when designing future studies on the same subject.